

# Borneo coral reefs subject to high sediment loads show evidence of resilience to various environmental stressors

Nicola Browne[1,2], Christina Braoun[3], Jennifer McIlwain[1,2], Ramasamy Nagarajan[4] and Jens Zinke[1,2,3,5,6]

[1] Molecular and Life Sciences, Curtin University, Perth, WA, Australia
[2] Curtin Malaysia Research Institute, Curtin University, Miri, Sarawak, Malaysia
[3] Department of Biology, Chemistry and Pharmacy, Freie Universität, Berlin, Germany
[4] Department of Applied Geology, Curtin University, Miri, Sarawak, Malaysia
[5] School of Geography, Geology and Environment, Centre for Palaeobiology, University of Leicester, Leicester, UK
[6] Australian Institute of Marine Science, Townsville, WA, Australia

Corresponding author
Jennifer McIlwain,
jennifer.mcilwain@curtin.edu.au

## ABSTRACT

For reefs in South East Asia the synergistic effects of rapid land development, insufficient environmental policies and a lack of enforcement has led to poor water quality and compromised coral health from increased sediment and pollution. Those inshore turbid coral reefs, subject to significant sediment inputs, may also inherit some resilience to the effects of thermal stress and coral bleaching. We studied the inshore turbid reefs near Miri, in northwest Borneo through a comprehensive assessment of coral cover and health in addition to quantifying sediment-related parameters. Although Miri's Reefs had comparatively low coral species diversity, dominated by massive and encrusting forms of *Diploastrea*, *Porites*, *Montipora*, *Favites*, *Dipsastrea* and *Pachyseris*, they were characterized by a healthy cover ranging from 22 to 39%. We found a strong inshore to offshore gradient in hard coral cover, diversity and community composition as a direct result of spatial differences in sediment at distances <10 km. As well as distance to shore, we included other environmental variables like reef depth and sediment trap accumulation and particle size that explained 62.5% of variation in benthic composition among sites. Miri's reefs showed little evidence of coral disease and relatively low prevalence of compromised health signs including bleaching (6.7%), bioerosion (6.6%), pigmentation response (2.2%), scars (1.1%) and excessive mucus production (0.5%). Tagged colonies of *Diploastrea* and *Pachyseris* suffering partial bleaching in 2016 had fully (90–100%) recovered the following year. There were, however, seasonal differences in bioerosion rates, which increased five-fold after the 2017 wet season. Differences in measures of coral physiology, like that of symbiont density and chlorophyll *a* for *Montipora*, *Pachyseris* and *Acropora*, were not detected among sites. We conclude that Miri's reefs may be in a temporally stable state given minimal recently dead coral and a limited decline in coral cover over the last two decades. This study provides further evidence that turbid coral reefs exposed to seasonally elevated sediment loads can exhibit relatively high coral cover and be resilient to disease and elevated sea surface temperatures.

# INTRODUCTION

Turbid reefs are commonly regarded as marginal reefs living near their environmental limits (*Kleypas, McManus & Menez, 1999*; *Guinotte, Buddemeier & Kleypas, 2003*; *Perry & Larcombe, 2003*; *Palmer et al., 2010*; *Goodkin et al., 2011*). As such, these reefs are traditionally perceived to be in a reduced health status (*Kleypas, 1996*; *Kleypas, McManus & Menez, 1999*) and more sensitive to rising sea surface temperatures (SST; *Nugues & Roberts, 2003*; *Crabbe & Smith, 2005*; *Fabricius, 2005*; *Woolridge, 2008*). Yet, there is growing evidence that turbid reefs may actually be more resilient to future climate change effects (*Goodkin et al., 2011*; *Morgan et al., 2017*) and serve as refugia for surviving corals (*Cacciapaglia & van Woesik, 2015*, *2016*; *Morgan et al., 2016*). This has been demonstrated on turbid reefs that experience significant sediment and nutrient inputs, yet are still characterized by high coral cover, low bleaching and rapid recovery rates from bleaching and cyclonic events (*Larcombe, Costen & Woolfe, 2001*; *Browne, Smithers & Perry, 2010*; *Richards et al., 2015*; *Morgan et al., 2016*). Studying the level of resilience and survival of turbid reefs in different environmental settings will provide clearer insights into the future of reefs subject to climate change (*Guinotte, Buddemeier & Kleypas, 2003*; *Hennige et al., 2010*; *Richards et al., 2015*).

Despite elevated resilience to naturally turbid conditions, many inshore turbid reefs face threats from local pressures, largely related to declining water quality and increased sediment input. In South East (SE) Asia, 95% of reefs are threatened from local sources and, therefore, are regarded as the most endangered reefs globally (*Burke et al., 2011*). From the 1980s to early 2000s hard coral cover (HCC) on reefs in SE Asia declined from 45% to 22% at an average rate of 2% loss per year (*Bruno & Selig, 2007*). Most SE Asian reefs are located in close proximity to countries with rapidly emerging economies and fast population growth (*Wilkinson, 2006*; *Burke et al., 2011*; *Heery et al., 2018*). They are further characterized by poorly developed environmental policies, inadequate regulation, lack of enforcement, a shortage of institutional and technical capacity, insufficient community support and involvement, and conflicts and tensions between stakeholders (*Fidelman et al., 2012*). The synergistic effects of these factors have led to poor water quality on many inshore reefs via pollution and sediment input derived by rapid land development, and over-fishing activities (*McManus, 1997*; *Wilkinson, 2006*). As a consequence, sedimentation rates are high with SE Asian coastal systems experiencing the highest siltation loads globally (*Kamp-Nielsen et al., 2002*; *Syvitski et al., 2005*).

Nearshore coral reefs along the north central section of Sarawak, on the island of Borneo, are highly diverse with an estimated 518 fish species (*Shabdin, 2014*) and 203 hard coral species from 66 genera (*Elcee Instrumentation Sdn Bhd, 2002*). Sarawak is a deforestation hotspot with only 3% of its forest cover intact (*Bryan et al., 2013*). Ongoing deforestation and poor land use practices are a growing threat for these biological diverse reefs that also support local fisheries and an expanding dive tourism industry

(*Elcee Instrumentation Sdn Bhd, 2002*). As such, in 2007 a marine park (the Miri-Sibuti Coral Reef National Park, MSCRNP) that covered 11,020 km$^2$ was established to promote and protect 30 coral reefs adjacent to Miri, the second largest town in Sarawak. In 2001, a broad assessment of coral reef health within the park indicated that live coral cover was approximately 35–50% and recently dead coral cover was 0.5% (*Elcee Instrumentation Sdn Bhd, 2002*). Subsequent Reef Check surveys in 2010 and 2014 concluded these same reefs were experiencing multiple stressors, but were in 'fair' condition (~40% HCC; *Reef Check Malaysia, 2010*, *2014*). However, despite these surveys, there is limited quantitative data on coral health and biodiversity (*Shabdin, 2014*), and more importantly no comprehensive assessment of environmental drivers of reef health. For example, the Baram River (10 km north of the reef complex), is known to discharge 2.4 x 10$^{10}$ kg yr$^{-1}$ of sediments into the coastal zone (*Nagarajan et al., 2015*), such that sediment and nutrient influx are considered to be the greatest threat to these poorly studied reefs (*Pilcher & Cabanban, 2000*; *Ferner, 2013*; *Shabdin, 2014*). Without thoroughly quantifying sediment impacts on corals, no conclusions can be made on coral tolerance levels, the drivers of community composition and future resilience to both local and global pressures. Given the Baram River delta is in a destructive phase due to rising sea level (*Lambiase, bin Abdul Rahim & Peng, 2002*), together with the increased frequency and intensity of rainfall events and plans for future modification of both the river and adjacent land development (*Nagarajan et al., 2015*), it is likely that threats from sediments will only increase.

The reefs within the MSCRNP provide a valuable opportunity to address several knowledge gaps on turbid coral reef health and their potential resilience to local and global threats. The last comprehensive assessment of coral cover on Miri's reefs was in 2001 (*Elcee Instrumentation Sdn Bhd, 2002*), with no assessments of coral taxa health and disease for any Sarawak reef recorded to date. In particular, coral disease studies are rarely undertaken on SE Asian reefs largely due to a lack of resources and expertise (*Green & Bruckner, 2000*; *Raymundo et al., 2005*; *Heintz, Haapkylä & Gilbert, 2015*). The lack of quantitative data on the health and stability (as defined by resistance, resilience and maintenance of key functional groups) of these reefs coupled with ongoing unsustainable land use practices in Sarawak, raises concerns over their long-term viability. This is of particular concern as Sarawak reefs have been estimated to provide six million AUD per year in tourism and 13.5 million AUD from fisheries (*Elcee Instrumentation Sdn Bhd, 2002*). We argue there is an urgent need for a comprehensive assessment of coral cover and health measured alongside key environmental and sediment-related parameters. The key objectives of this study therefore are to: (1) quantify benthic cover and health, (2) compare the prevalence of impaired health in the dominant coral species and (3) identify key parameters related to sediment delivery that are associated with benthic cover and health along an inshore to offshore gradient. These data will improve our understanding of turbid coral reefs composition and potential resilience to both local and global stressors, and promote current management strategies that aim to protect inshore turbid reefs from future changes to land use.

## MATERIALS AND METHODS

### Study sites

The study was conducted on three low profile submerged patch reefs (Eve's Garden, Anemone Garden and Siwa Reef) in the MSCRNP (Fig. 1). These sites were of a comparable depth (5–15 m) and size (<0.11 km$^2$), and benthic composition over the reefs are comparatively homogenous. Eve's Garden (EG) is a shallow inshore reef close to shore (7.3 km) with a coral community dominated by plate and massive corals such as *Pachyseris* sp. and *Porites* sp. (Ferner, 2013). Anemone Garden (AG) is further offshore (11.7 km) and consists of a considerable density of anemone colonies, with plating forms of *Acropora* sp. and exceptionally large massive *Porites* sp. and *Diploastrea* sp. colonies (one to five m length). Siwa Reef (SW) situated further to the south is the most biologically diverse of the studied reefs consisting of encrusting and massive coral forms (Ferner, 2013). These reefs lie on an inshore to offshore transect from the Baram (sediment influx $2.4 \times 10^{10}$ kg . year$^{-1}$; Nagarajan et al., 2015) and Miri River mouths, located to the north of EG (10 km from Miri river and 30 km from Baram river).

Physical (temperature, light, turbidity and sediment trap accumulation) and biological (benthic cover, coral health) data were collected at the end of the dry season (15th September to 20th October 2016) and during the wet season (11th May to 3rd June 2017). At each of the three reefs, six replicate line transects (20 m), separated by 20 m intervals to ensure independence were run across the reef surface (EG = 8–12 m; AG =10–14 m; SW = 8–14 m). These reefs are not characterized by typical windward and leeward reef edges, and zonation patterns but are low profile patch reefs where the majority of the reef sits in one relatively flat plane, sloping gently on all sides to the sea floor. As such, all transects were laid out along the same axis across the flat section of each reef.

### Physical data collection

Seasonal changes in light (measured with Photosynthetic Irradiance Recording System by Odyssey, New Zealand) and temperature (measured with HOBO Pro V2 loggers, Australia) were recorded every 10 min from September 2016 for 9 months (temperature at EG and AG) and 12 months (light at EG). To capture changes in suspended sediment loads over a tidal cycle, turbidity loggers were deployed (in a horizontal position) for two weeks at the end of the 2016 dry season (September; EG and SW) and end of the 2017 wet season (May; EG; AQUA logger 210/310TY, Aquatech, UK). Data on cloud cover, rainfall and wind speed from October 2016 to October 2017 were retrieved from the online database *World Wide Weather (2017)*.

To assess small-scale spatial variation in sediment trap accumulation, four sediment traps per three transects (eight traps in total per reef) were deployed at each reef in September 2016 to collect sediments during the NE monsoon. Each trap consisted of three cylindrical PVC plastic containers (diameter of 7.6 cm by 15 cm high) attached to a metal rod positioned 30 cm above the substrate (Storlazzi, Field & Bothner, 2011). The traps remained in-situ until May 2017. To determine if trapped sediments were from local

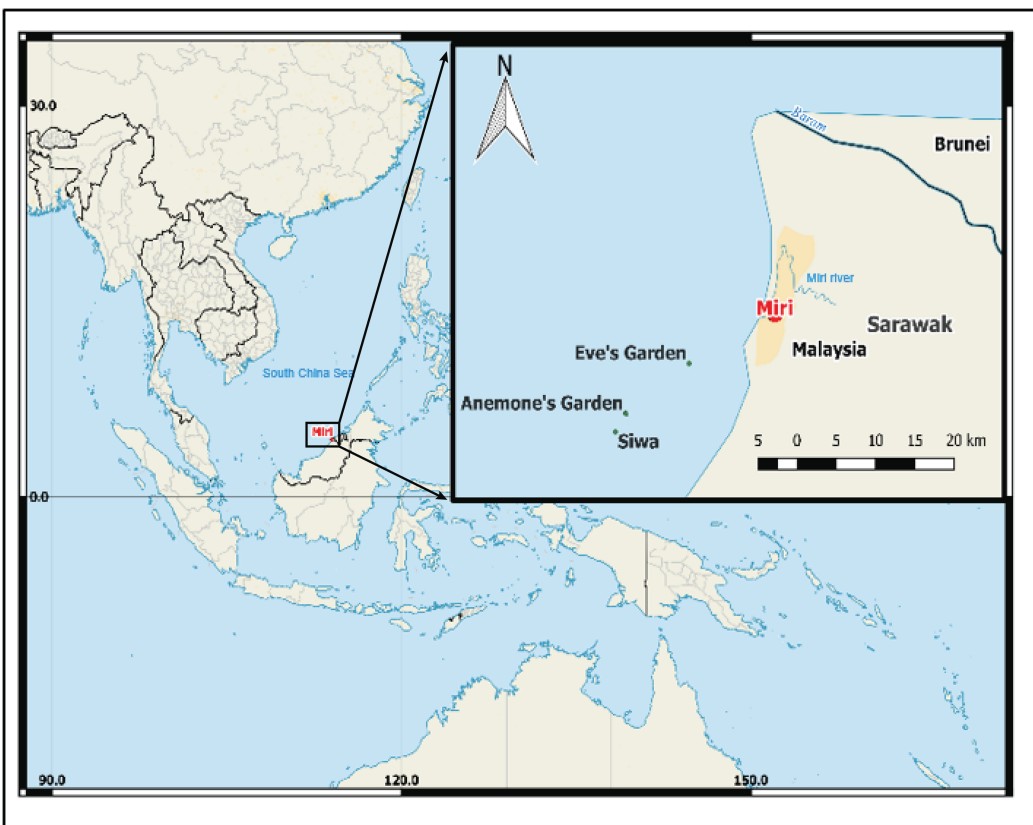

**Figure 1 Map of south China Sea with enlarged map of study area, showing locations of the three reefs, Miri city and the closest rivers.** (Image credit Hedwig Krawczyk modified from Natural Earth – Free vector and raster map data).

resuspension or transported on to the reef, 500 g of benthic sediment at the base of each trap was sampled. The content of each trap was emptied into a labelled plastic bag and stored at −20 °C until further analysis at the Curtin University Malaysia Laboratory facilities (Laboratory SK2 204).

Sediment samples were analyzed for weight and particle size characteristics. Frozen samples were thawed and allowed to settle overnight. Water remaining on the surface was filtered (0.45 nm filter paper) to capture the fine suspended sediments. The sediments (washed, settled and filtered) were oven-dried at 60 °C for 2–3 weeks and weighed to the nearest 0.001 g. Sedimentation accumulation rate (g cm$^2$ day$^{-1}$) was calculated as the weight of sediment trapped (g) divided by the number of days the trap was deployed and the surface area of the trap (cm$^2$). For the grain size analysis, the settled dry sediments were manually homogenized and weighed before sieving. The sediments were homogenized using a pestle and mortar given the sediments were mostly sand and loosely aggregated. Sediments were separated into five class fractions (>1 mm, 500 to <1,000 μm, 250 to <500 μm, 125 to <250 μm and 63–125 μm) by placing the sieve stack on a mechanical shaker for 20 min. Each of the five sediment fractions was weighed to the nearest 0.001 g.

## Biological data collection

### In water data collection

The benthic cover and coral diversity (to genus level) in September 2016 were assessed using the photographic transect method (*Bégin, Wurzbacher & Côté, 2013*). Photographs were taken using a Canon Powershot G7 mark II digital camera at a fixed height of 0.75 m above the transect line every 1 m along the transect ($n = 21$). Photographs (0.5 m$^2$) were analyzed in Coral Point Count (CPCe) with a uniform grid of 25 points to calculate benthic cover for each of eight categories (hard coral, soft coral, recently dead coral, turf algae, macroalgae, crustose coralline algae, sponge and abiotic substrate) (CPCe; *Kohler & Gill, 2006*). Recently dead coral (<1 year) was considered to include dead coral that had visible corallites and limited algae growth. The hard coral category was further subdivided into 38 genera common to the Indo-Pacific region according to *Kelley (2009)*.

To assess seasonal fluctuations in coral reef health, signs of compromised health (disease, bleaching, bioerosion, pigmentation response, excessive mucus production and scars) were recorded in September 2016 and May 2017. The belt transect methodology was used to cover a wider area along the transect line via a zig-zag pattern 1 m either side of the transect line (40 m$^2$ for each 20 m transect). Coral colonies within each belt transect were identified to genus level and classified as either healthy or affected by an impaired health sign (*Beeden et al., 2008*). Signs of bioerosion included the presence of organisms such as Christmas tree worms, boring bivalves and sponges, and bleaching was further subdivided into whole, partial, focal and non-focal bleaching (as defined in *Beeden et al., 2008*). To determine if bleached corals recovered or died, 14 coral colonies from EG and SW in both sampling seasons that showed signs of bleaching were tagged and photographed (Four *Diploastrea* sp., Six *Pachyseris* sp., Four *Porites* sp.). The percentage of bleached tissue was assessed from scaled photographs using CPCe software (1 = normal, 2 = pale, 3 = 0–20%, 4 = 20–50%, 5 = 50–80% and 6 = 80+% bleached). While this is a low sample size, the data are included to provide further insight into the recovery potential of corals on these reefs.

### Symbiont density and chlorophyll a analysis

In May 2017, fragments of three coral genera (*Montipora* sp., *Pachyseris* sp. and *Acropora* sp.) were collected from EG, AG and SW for chlorophyll *a* and symbiont density analysis. Higher chlorophyll *a* and symbiont densities are typically recorded on turbid reefs (*Browne et al., 2015*) as this increases the coral's ability to photosynthesis under low light levels as they acclimate to suspended sediments (*Hennige et al., 2010*). Fragments (5–10 cm for branching corals and ~10 x 10 cm for foliose corals) were collected using cutters and placed in plastic bags. Samples were placed on ice during transportation back to the laboratory where they were stored at −80 °C until further analysis. Symbiont density and chlorophyll *a* content were quantified following the removal of coral tissue from the skeleton. The protocol for extracting tissue was adapted from *Ben-Haim, Zicherman-Keren & Rosenberg (2003)* (Supplementary Material).

## Statistical analysis

Univariate statistical analysis was conducted in R Studio Desktop version 1.1.383. Prior to analysis, normal distribution and homogeneity of variances were checked using the Shapiro–Wilk test and the Levene's test, respectively. To assess if there were significant differences in benthic cover (hard coral, soft coral, algae) and diversity among sites a one-way analysis of variance (ANOVA, $n = 6$, $\alpha = 0.05$) was used followed by a Tukey HSD post-hoc test (Bonferroni method), if necessary. Significant differences in the prevalence of compromised health signs (bleaching, bioerosion, mucus production, pigmentation and scars) among sites and between seasons were identified for both total HCC and for the most abundant coral genera (*Porites*, *Pachyseris*, *Montipora*, *Diploastrea*, *Acropora*) using a Full Factorial ANOVA (FF ANOVA, $n = 6$, $\alpha = 0.05$) and a Tukey HSD post-hoc test. If required, a log10 transformation was carried out for datasets to meet homogeneity of variance. However, as the bleaching recovery was assessed using a scale, these data were tested using a Wilcoxon test to determine if there had been a significant recovery in tagged coral colonies between years. To determine differences in physiology (chlorophyll *a* content and zooxanthellae density) between the three coral genera sampled (*Acropora* $n = 17$, *Pachyseris* $n = 13$, *Montipora* $n = 15$) and across sites, a non-parametric Kruskal–Wallis test was performed. Furthermore, to evaluate cell health differences between the three genera and among reefs, the percentage of cells from each grade were compared using the Kruskal–Wallis test. Differences in sediment trap accumulation rates were tested among reefs (Kruskal–Wallis). In addition particle size characteristics (median, fine/course fraction) among reefs, and between the trapped sediments and the benthic sediments were also tested (FF ANOVA, $n = 18$).

Permutational multivariate analysis was conducted in PRIMER-7 version 7.0.13. A Distance-based Linear Model (DISTLM) was used to determine how much of the variation in community assemblage (hard coral cover = HCC, soft coral cover = SCC, algae, recently dead coral cover = DCC, H' index, number of coral genera) among transects and reefs was driven by depth, distance from the two nearby river mouths, distance from shore and differences in sediment trap accumulation rates and particle size characteristics. Depth was included in the analysis as depth is known to influence sediment dynamics (*Wolanski et al., 2005*) as well as declines in light associated with suspended sediments (*Falkowski, Jokiel & Kinzie, 1990*). A distance-based resemblance matrix was created for the biological data set using Bray–Curtis similarity values following a square-root transformation and for the environmental data using Euclidean distances and normalized values. A DISTLM, using the BEST fit model with the Akaike's Information Criterion (AIC) and 9,999 permutations was performed using the resemblance matrices. The multivariate scale relationship between the predictor (environmental) and response variables (biological) was presented on a plot with a distance-based redundancy analysis (dbRDA; *Legendre & Anderson, 1999*). To investigate whether environmental factors contributed to differences in health status among sites again a DISTLM model was used followed by dbRDA plotting as above. Predictor variables included substrate structure (HCC, diversity) and physical conditions (depth, sediment trap accumulation rate,

particle size characteristics, distance from both river mouths and distance from shore). Hard coral cover and diversity were used since higher HCC can contribute to a greater probability of impaired coral health (*Bruno & Selig, 2007*). In contrast, reefs that are more diverse can lower susceptibility as it reduces the quick spread of a disease (*Raymundo et al., 2005*; *Aeby et al., 2011*). As sediment data were obtained at the end of the wet season (May 2017), these were used to explain the 2017 health data. For the 2016 coral health data, which had no associated sediment data, only sampling year, HCC and coral diversity together with distance from shore and rivers were used as explanatory variables.

## RESULTS

### Physical parameters

The dry season was characterized by less variable, warmer SST (mean monthly range = 30.0–30.7 °C; Fig. S2), greater in-water light penetration (mean monthly range at EG = 156–320 µmol photons $m^{-2}$ $s^{-1}$) and reduced rainfall (mean monthly rainfall range = 78–166 mm) and cloud cover (Fig. 2). In contrast, the wet season was cooler (mean monthly range = 28–30.1 °C) with higher rainfall (mean monthly range 126–234 mm) and reduced light levels on the reef (mean monthly range at EG = 19–150 µmol photons $m^{-2}$ $s^{-1}$). Wind speeds were also slightly elevated during the wet season months (Fig. 2D). Mean sediment trap accumulation rates following the wet season ranged from 13 to 28 mg $cm^{-2}$ $day^{-1}$, with a rate almost three times higher at EG compared to AG and SW ($H_2$ = 10.3, $p < 0.005$; Fig. 3). Site differences in potential sediment load were also observed during the dry season with higher and more variable turbidity recorded at the nearshore EG reef (mean monthly range = <1–24 FTU) than at SW (mean monthly range = 1–7 FTU) located 10 km further south from the large Baram river mouth (Fig. S3).

All three reefs were dominated by sand (>98%), with the median particle size of benthic sediments significantly increasing ($F_2$ = 13.6, $p < 0.005$) with distance from the mouths of the Baram and Miri rivers (Fig. 4). Benthic sediments at SW comprised 58% of very coarse sand, nearly three times that of EG (20%) ($F_3$ = 24.9, $p < 0.001$; PH: SW > EG, AG; Fig. S4) and a significantly smaller proportion of medium/fine sands ($F_2$ = 17.2, $p$ = <0.001; PH: SW > AG > EG). In contrast there was little difference in the median particle size from the sediment traps among sites ($F_2$ = 2.25, $p$ = 0.133), although particle sizes of the benthic sediment were significantly greater compared to the trapped sediments ($F_1$ = 60.93, $p < 0.001$).

### Benthic cover

Hard coral cover increased with distance from the major sediment source (Baram River) and varied significantly among sites ($F_2$ = 5.3, $p$ = 0.01; PH: SW > EG). Siwa Reef had the highest HCC (39.3%) and EG almost half the HCC (21.9%; Fig. 5). Soft corals also varied significantly but declined with increasing distance from the major sediment source ($H_2$ = 8.6, $p$ = 0.01; MWPH: EG > AG, SW) with EG having nearly 15-fold higher cover than SW. Turf algae dominated the algal community and contributed to 52–57% of all

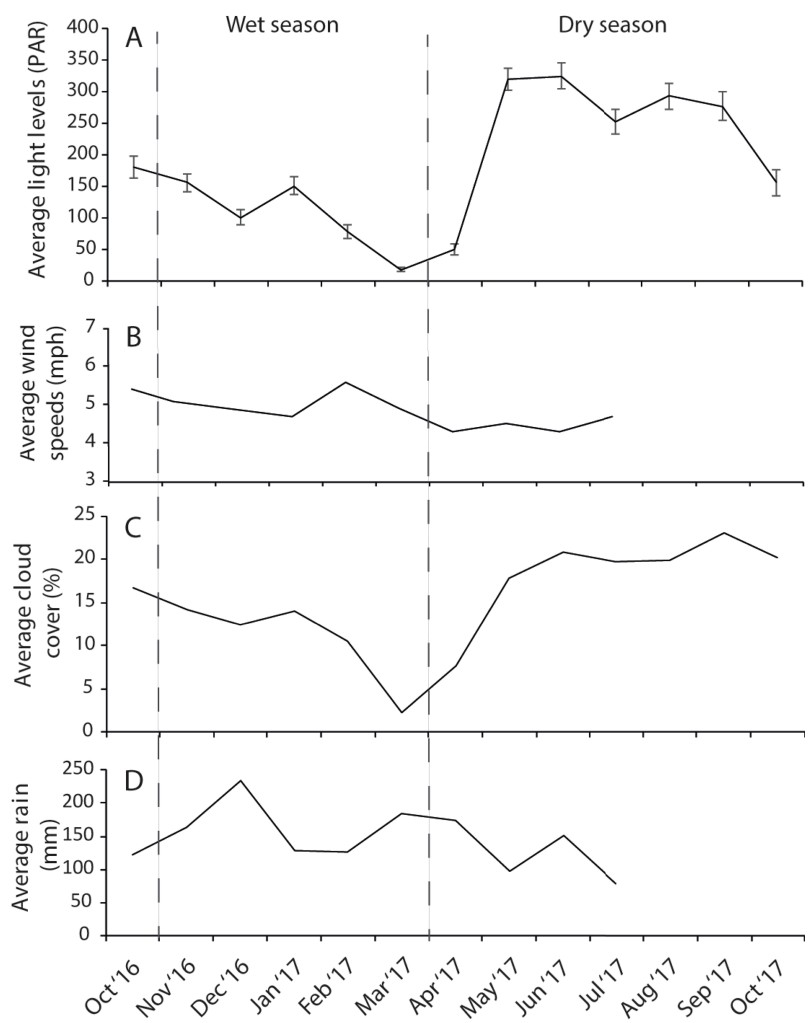

**Figure 2 Average monthly data for (A) light, (B) wind speeds, (C) cloud cover, and (D) rain fall.** Light data were collected at EG as part of this study whereas wind, cloud and cover data were taken from the worldwideweatheronline.com website (error bars = SE).

reefs' benthos. However, there was no significant difference in turf algal cover among reefs ($F_2 = 0.103$, $p > 0.05$). Recently dead coral cover was consistently low among sites (4.25%) as was crustose coralline algae, which was typically covered in turf algae.

In total 28 genera were recorded (Table 1). Coral diversity was considerably different among sites ($F_2 = 4.6$, $p = 0.03$; PH: SW > EG) with SW having the highest richness and 25 genera (H′ = 1.93), and EG and AG with 16 and 14 genera, respectively (H′ ~1.4). The surveyed sites were composed of similar communities, with the most dominant genera including *Diploastrea* sp., *Porites* sp., *Montipora* sp., *Favites* sp., *Dipsastrea* sp. and *Pachyseris* sp. (Table 1). All other species comprised a small fraction of the community (<2% cover). Most notable differences in the composition were with the high cover of *Diploastrea* sp. at AG and EG, *Galaxea* sp. at EG and *Acropora* and *Montipora* sp. at SW.

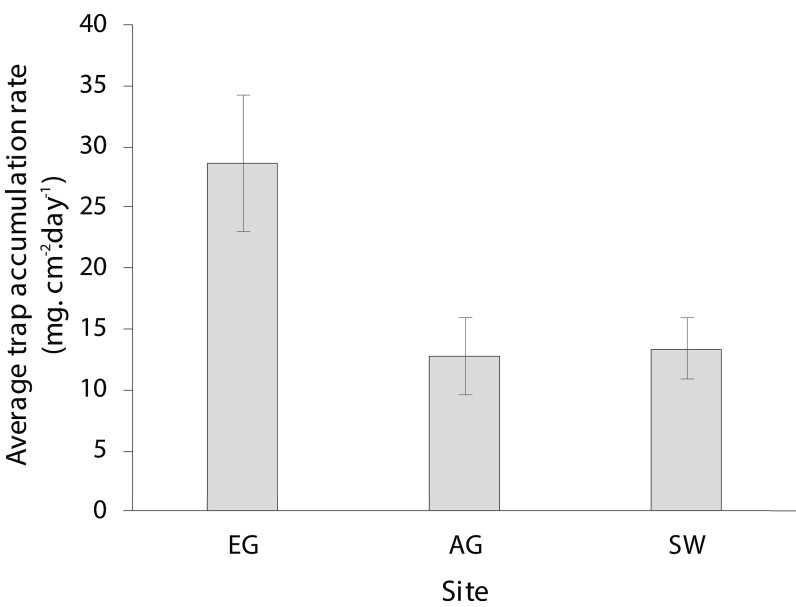

**Figure 3 Average sedimentation rates at the three surveyed sites (error bars = SE).**

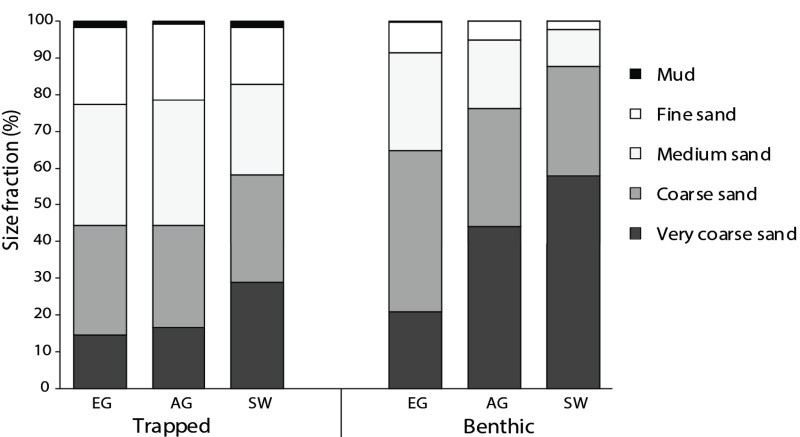

**Figure 4 Particle size data from the sediment traps and the benthos at EG, AG and SW.**

## Coral reef health

Of the compromised health signs recorded at each reef, the five most commonly observed were mucus production (0.5 + 0.3%), pigmentation response (2.2 + 0.7%), bioerosion (6.6 + 2%), bleaching (6.7 + 0.9%) and scars (1.1 + 0.4%; Fig. 6). No diseases per se were observed except at EG where one colony of massive *Porites* sp. had ulcerative white spots. Despite a decline in the prevalence of compromised health along an inshore to offshore gradient following the dry season in 2016, total prevalence (sum of the five commonly observed signs) was not statistically significant among sites and seasons ($p > 0.05$; Table 2). The prevalence of mucus production by corals at Eves Garden (5%), however, was nearly five times that of other reefs ($F_2 = 3.6$; $p < 0.05$; EG < AG, SW), and

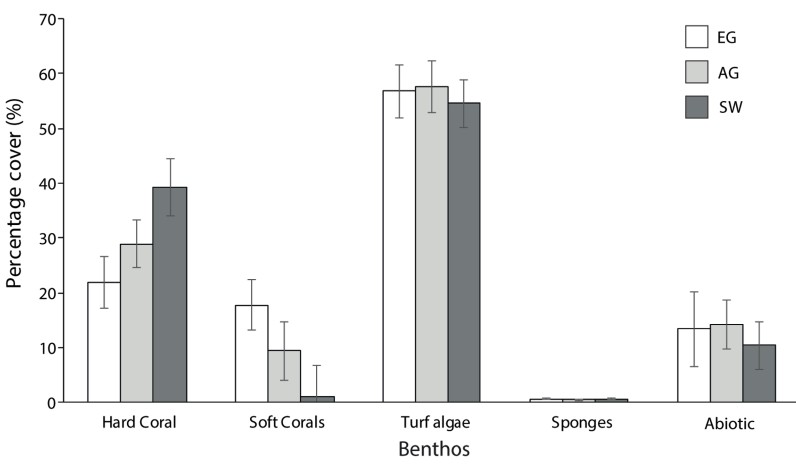

**Figure 5 Average percentage benthic cover at EG, AG and SW.** Sites are organizsed from inshore to offshore (error bars = SE).

SW recorded the lowest levels of pigmentation prevalence (Table 2; $F_2 = 5.3$; $p < 0.05$; AG > SW). In contrast, bioerosion was comparatively similar among sites within each season, but increased five-fold from 2.7 + 0.6% to 10 +1.3% following the 2017 wet season (Table 1; $F = 20.2$; $p < 0.001$; 2017 > 2016; Fig. 7). During both seasons, overall bleaching prevalence was ≤10% with partially bleached the most common form and whole bleaching the least observed (Fig. S5). Bleaching prevalence declined from 8.1 + 1.4% following the dry season to 5.4 + 1.1 % after the wet season. Although this decline was not statistically significant ($F = 3.3$; $p = 0.08$), the recovery of bleached corals that had been tagged the year before was significant (p = 0.002). The average bleaching scale dropped from 3.9 + 0.4 to 1.6 + 0.2 (Fig. 8) with all *Diploastrea* sp. and *Pachyseris* sp. colonies recovered by 90–100% in 2017.

Patterns of compromised health differed among five representative coral genera (*Acropora* sp., *Montipora* sp., *Pachyseris* sp., *Diploastrea* sp. and *Porites* sp.). *Acropora* sp. displayed the least signs of stress in both seasons (<3.5%). *Porites* sp. were the most compromised (2016 = 50.8 + 6%; 2017 = 72 + 5%; Fig. 9) and the only coral genera with a significant increase in stress symptoms ($p = 0.004$), because of a 40% increase in bioerosion after the wet season ($F_1 = 10.17$; p < 0.001; Table 3). *Montipora* sp. and *Diploastrea* sp. also suffered from an increase in bioerosion between sampling seasons, although this was not statistically significant ($p > 0.05$; Table 3). Despite a slight increase in the number of bleached *Porites* sp. corals, bleaching occurrence for the other four corals declined, most notably for *Pachyseris* sp. (55% to 3%; $F_1 = 9.03$; $p = 0.008$). Furthermore, the most abundant genus *Porites* sp. was the only coral to show elevated signs of pigmentation (>10%) although this health sign was less prevalent at SW, the most offshore site ($F_2 = 5.3$; $p = 0.01$; Table 3).

For the three coral genera, *Montipora* sp., *Pachyseris* sp. and *Acropora* sp., there was no significant difference in symbiont density ($H = 4.0397$, df = 2, $p > 0.05$) and chlorophyll *a* among sites ($H = 2.3769$, $p > 0.05$) although SW scored the highest of both measures ($3.2 \times 10^6$ + 5.5 cells/cm$^2$; 4.94 + 0.75 µg.cm$^2$; Fig. 10). Symbiont density differed among the three coral genera (chi-square = 23.1, df = 2, $p < 0.001$; MWPH: AC > MT, PH) with

**Table 1 Average (%) coral cover of the 28 genera observed at the three surveyed reefs illustrating the 10 most dominant coral genus.**

| Genus | Eve's Garden | Anemone's Garden | Siwa reef |
|---|---|---|---|
| *Acropora* (branching) | 0.07 ± 0.07 | | 2.60 ± 0.40 |
| *Diploastrea* (massive) | 14.80 ± 1.60 | 10.60 ± 3.70 | 0.40 ± 0.10 |
| *Echinopora* (encrusting) | | 0.50 ± 0.14 | 1.90 ± 1.60 |
| *Dipsastrea* | 0.90 ± 0.30 | 3.44 ± 0.40 | 3.60 ± 2.00 |
| *Favites* (massive) | 1.70 ± 0.80 | 2.40 ± 0.86 | 5.10 ± 1.60 |
| *Galaxea* | 3.00 ± 1 | 0.62 ± 0.20 | 0.90 ± 0.30 |
| *Merulina* | 1.60 ± 1.5 | 0.10 ± 0.03 | 1.33 ± 0.80 |
| *Montipora* (plate) | 1.30 ± 100 | 2.09 ± 1.10 | 8.60 ± 3.00 |
| *Pachyseris* (plate) | 2.10 ± 1.10 | 0.50 ± 0.30 | 2.00 ± 1.30 |
| *Porites* (massive/plate) | 5.70 ± 2.80 | 7.30 ± 1.50 | 7.30 ± 2.30 |
| *Astreopora* | | | 0.90 ± 0.60 |
| *Caulastrea* | | 0.07 ± 0.19 | 0.04 ± 0.04 |
| *Ctenactis* (solitary) | 0.07 ± 0.07 | 0.62 ± 0.15 | 0.14 ± 0.09 |
| *Echinophyllia* | 0.30 ± 0.30 | | 0.06 ± 0.06 |
| *Fungia* | | | 0.10 ± 0.01 |
| *Goniastrea* | | 0.10 ± 0.03 | 0.04 ± 0.04 |
| *Goniopora* | 0.03 ± 0.03 | | |
| *Heliofungia* | 0.10 ± 0.10 | | |
| *Leptoria* | 0.03 ± 0.03 | | 0.08 ± 0.08 |
| *Leptoseris* | 0.17 ± 0.17 | | 1.60 ± 1.50 |
| *Montastrea* | | | 0.04 ± 0.04 |
| *Oxypora* | 0.03 ± 0.03 | | 0.17 ± 0.17 |
| *Pectinia* | | | 0.08 ± 0.08 |
| *Physogyra* | | | 0.17 ± 0.17 |
| *Platygyra* (massive) | 0.90 ± 0.80 | 1.79 ± 1.60 | 0.60 ± 0.40 |
| *Psammocora* | 0.10 ± 0.10 | | |
| *Symphyllia* | | 0.40 ± 0.20 | 0.69 ± 0.30 |
| *Turbinaria* | | | 0.68 ± 0.68 |

*Acropora* sp. scoring four and five times higher symbiont densities (Fig. S6). Over 50% of the symbionts observed where healthy (stage 1; Fig. S7A) with slightly more healthy cells observed at SW ($H = 1.7$, $p > 0.05$) and marginally more degraded cells (stage 5) observed at AG ($H = 3.4$, $p > 0.05$). Among genera, *Acropora* had a greater number of healthy cells (69 + 3.9%) than both *Montipora* (49.4 + 5) and *Pachyseris* (52.6 + 4.8; $H = 14.4$, $p < 0.001$; Fig. S7B).

## Environmental associations with benthic cover and health

Environmental variables (depth, sediment trap accumulation rate, distances from shore/ river mouth, concentration of silt/fine/coarse particles, median particle size) explained 62.5% of the variation in benthic composition among reefs. Key drivers ($p < 0.05$) were

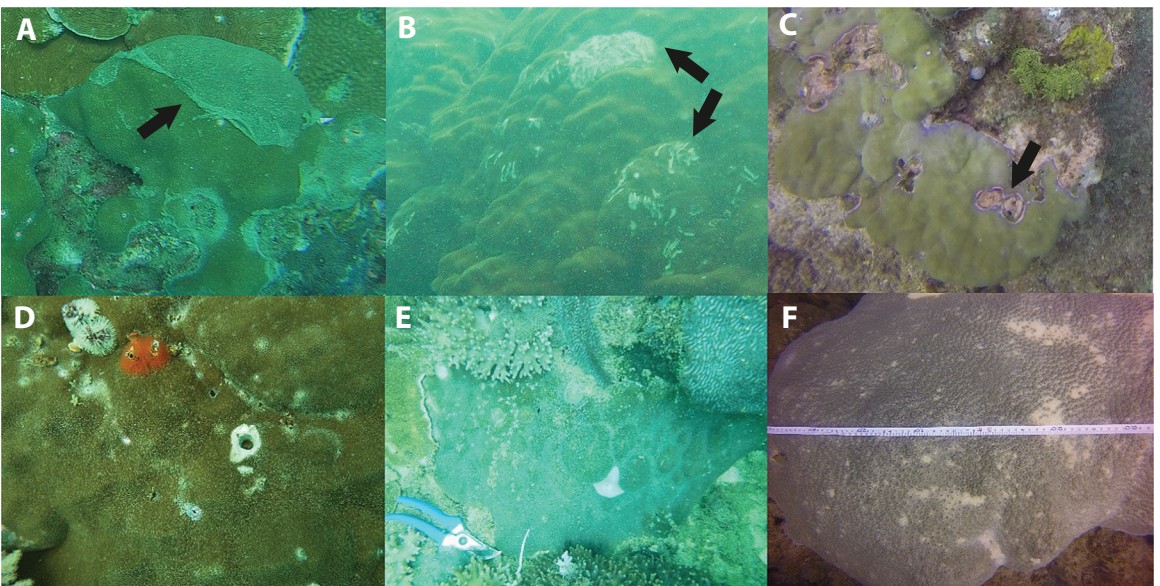

**Figure 6 Signs of impaired health.** (A) Excessive mucus and slothing, (B) Feeding scars from parrotfish, (C) pigmentation response in *Porites* sp. (D) Christmas tree worms and bivalves, (E) Non-focal bleaching, and (F) Partial bleaching. Black arrow points to associated impaired health sign.

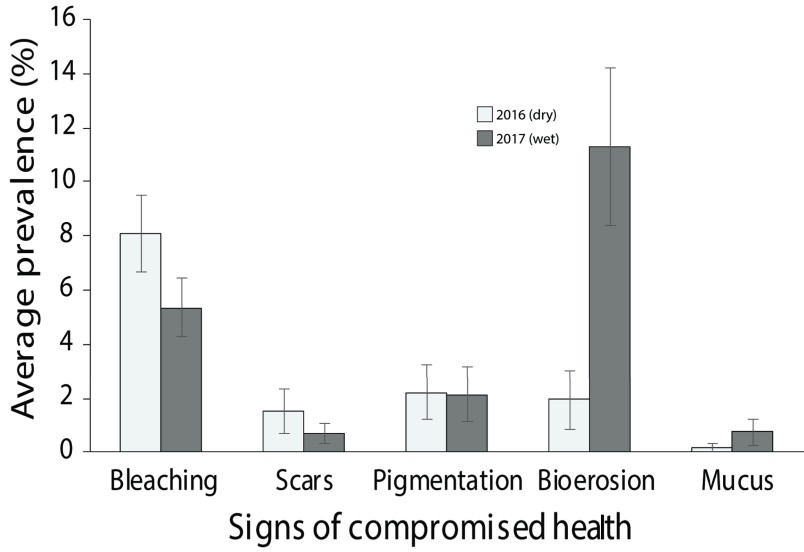

**Figure 7 Average prevalence of the dominant signs of impaired health across all three surveyed sites (EG, AG and SW) following the 2016 dry season and 2017 wet season (error bars = SE).**

distance from river mouth (30.3%) and shore (1%), median particle size (16.4%), and sediment trap accumulation rate (2.3%; Table 4). Variability among sites was higher than within, with sediment trap accumulation rate and particle size a key driver of benthos at EG and AG, and distance of river and shore more closely associated with SW (Fig. 11).

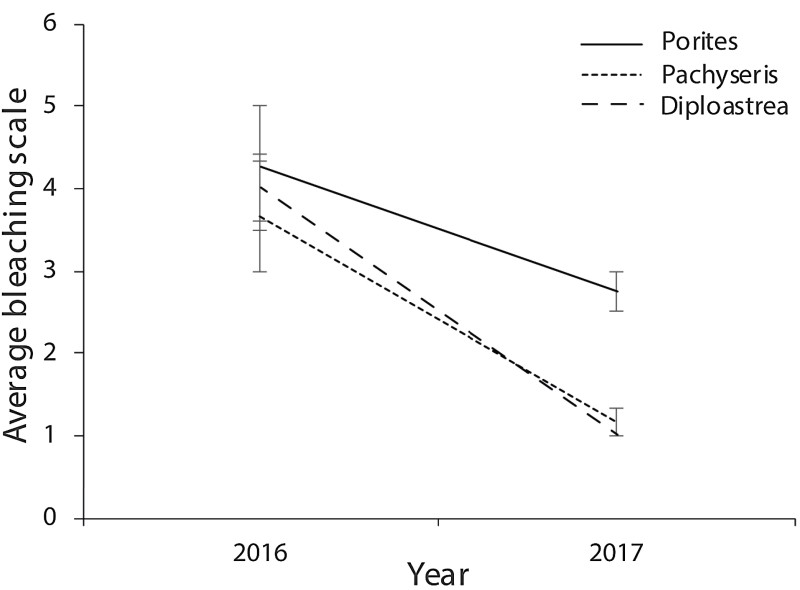

**Figure 8** Average bleaching scale for the three coral genus across the three survey sites (EG, AG and SW) that were tagged in September 2016 following the warm dry season and cooler wet season (error bars = SE). 1 = normal, 2 = pale, 3 = 0–20% bleached, 3 = 21–50% bleached, 4 = 51–80% bleached, 5 = 81–100% bleached.

**Table 2 Statistical results from two-way ANOVA of the total impaired health and each impaired health indicator with site (EG = Eves Garden, AG = Anenomes Garden, SW = Siwa) and season (2016, 2017), and the interaction.**

| Health sign | Factor | df | *F* value | *p* value | Post hoc |
|---|---|---|---|---|---|
| Total impaired health | Site | 2 | 0.25 | 0.780 | |
| | Season | 1 | 1.11 | 0.300 | |
| | Site*Season | 2 | 0.15 | 0.860 | |
| Bleaching | Site | 2 | 0.19 | 0.830 | |
| | Season | 1 | 3.30 | 0.080 | |
| | Site*Season | 2 | 0.69 | 0.510 | |
| Mucus | Site | 2 | 3.60 | *0.040* | EG < AG, SW |
| | Season | 1 | 0.15 | 0.700 | |
| | Site*Season | 2 | 7.20 | *0.003* | |
| Bioerosion | Site | 2 | 0.87 | 0.430 | |
| | Season | 1 | 20.20 | *<0.001* | 2017 > 2016 |
| | Site*Season | 2 | 3.80 | 0.040 | |
| Pigmentation | Site | 2 | 5.30 | *0.010* | AG > SW |
| | Season | 1 | 1.00 | 0.320 | |
| | Site*Season | 2 | 0.82 | 0.440 | |
| Scars | Site | 2 | 0.10 | 0.910 | |
| | Season | 1 | 0.33 | 0.570 | |
| | Site*Season | 2 | 2.59 | 0.090 | |

**Note:**
Bold text indicates significant difference.

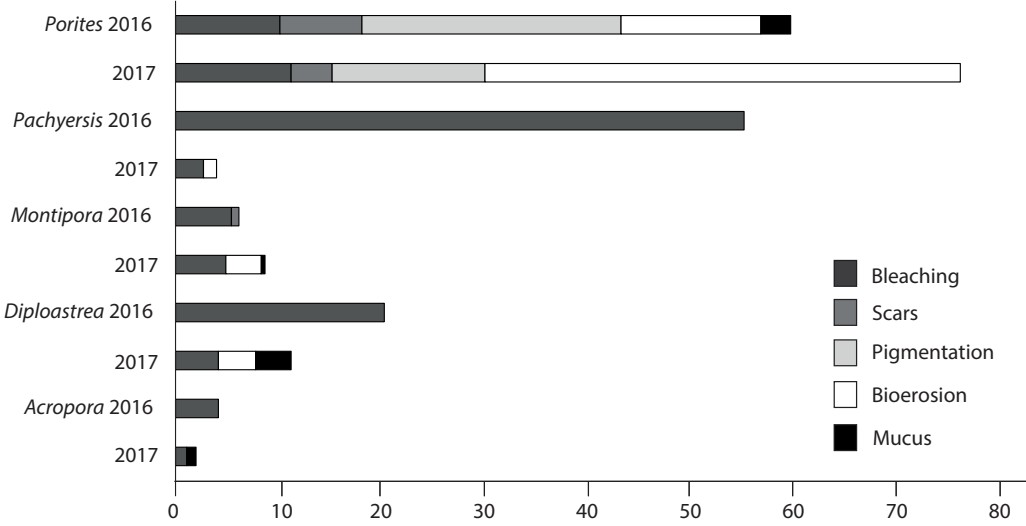

**Figure 9 Prevalence of the most common impaired health signs following the 2016 dry season and the 2017 wet season for the five most common observed coral genus across all three sites surveyed (EG, AG < SW).**

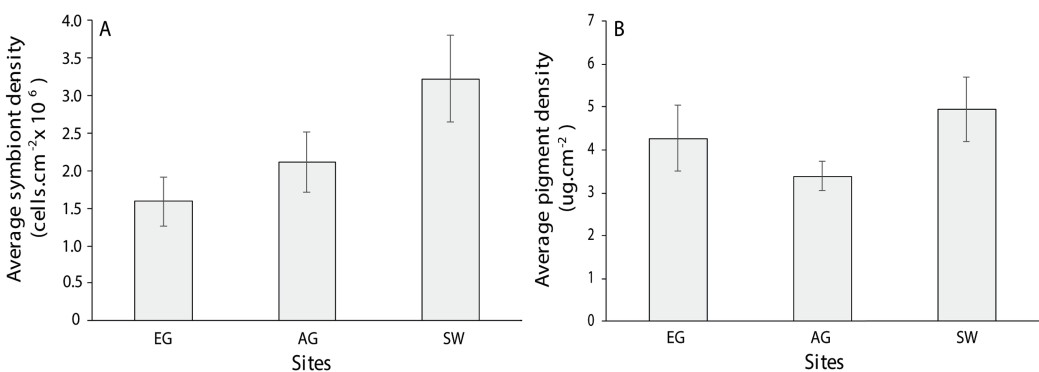

**Figure 10 Average symbiont density (A) and chlorophyll *a* pigment density (B) across the three coral species assessed (*Acropora, Monitpora* and *Pachyseris*) at EG, AG and SW (error bars = SE).**

To determine key drivers of coral health, two DistLM models were run. The first model included health data from both sampling seasons, with six explanatory variables (season, HCC, diversity, distance from river mouth and shore, and depth). The second model included health data and sediment-related variables following the wet season and sediment trap contents (sediment trap accumulation rate, concentration of silt/fine/course sediments, median particle size). For the first model, year, HCC and diversity significantly explained <31% of the variation in coral health among transects and sites (Table 5). Sites within a sampling season were separated along a HCC and diversity gradient (Fig. 12), with transects at SW typically characterized by higher HCC and diversity but lower prevalence of scars, pigmentation and bleaching (Fig. S8). Furthermore, repeat transects were separated between seasons, with those completed in 2017 recording higher bioerosion, but

**Table 3 Statistical results from two-way ANOVA of the total impaired health and each impaired health indicator for the five most dominant coral genera with site and season and the interaction.**

| Species | Health sign | Factor | df | *F* value | *p* value | Post hoc |
|---------|-------------|--------|----|-----------|-----------|----------|
| *Porites* | Total | Site | 2 | 1.71 | 0.202 | |
| | | Year | 1 | 10.17 | ***0.004*** | 2017 > 2016 |
| | | Site*year | 2 | 4.00 | ***0.031*** | |
| | Bleaching | Site | 2 | 0.36 | 0.701 | |
| | | Year | 1 | 0.08 | 0.774 | |
| | | Site*year | 2 | 1.81 | 0.185 | |
| | Mucus | Site | 2 | 6.72 | ***0.034*** | EG > SW |
| | | Year | 1 | 2.64 | 0.104 | |
| | | Site*year | | | | |
| | Bioerosion | Site | 2 | 1.61 | 0.219 | |
| | | Year | 1 | 21.79 | ***<0.001*** | 2017 > 2016 |
| | | Site*year | 2 | 6.29 | ***0.006*** | |
| | Pigmentation | Site | 2 | 8.79 | ***0.001*** | Eg, AG > SW |
| | | Year | 1 | 2.49 | 0.128 | |
| | | Site*year | 2 | 2.09 | 0.145 | |
| | Scars | Site | 2 | 0.46 | 0.637 | |
| | | Year | 1 | 0.38 | 0.543 | |
| | | Site*year | 2 | 2.25 | 0.126 | |
| *Pachyseris* | Total | Site | 2 | 0.30 | 0.744 | |
| | | Year | 1 | 9.02 | ***0.008*** | 2016 > 2017 |
| | | Site*year | 2 | 0.14 | 0.869 | |
| | Bleaching | Site | 2 | 0.37 | 0.699 | |
| | | Year | 1 | 9.69 | ***0.006*** | 2016 > 2017 |
| | | Site*year | 2 | 0.11 | 0.897 | |
| | Bioerosion | Site | 2 | 0.49 | 0.622 | |
| | | Year | 1 | 1.42 | 0.249 | |
| | | Site*year | 2 | 0.39 | 0.685 | |
| *Montipora* | Total | Site | 2 | 0.77 | 0.476 | |
| | | Year | 1 | 1.65 | 0.211 | |
| | | Site*year | 2 | 1.45 | 0.254 | |
| | Bleaching | Site | 2 | 2.06 | 0.149 | |
| | | Year | 1 | 0.29 | 0.594 | |
| | | Site*year | 2 | 0.73 | 0.494 | |
| | Bioerosion | Site | 2 | 0.83 | 0.449 | |
| | | Year | 1 | 0.83 | 0.371 | |
| | | Site*year | 2 | 0.68 | 0.519 | |
| *Diploastrea* | Total | Site | 2 | 0.66 | 0.527 | |
| | | Year | 1 | 0.10 | 0.752 | |
| | | Site*year | 2 | 2.54 | 0.104 | |
| | Bleaching | Site | 2 | 0.63 | 0.541 | |
| | | Year | 1 | 1.69 | 0.209 | |

| Species | Health sign | Factor | df | F value | p value | Post hoc |
|---|---|---|---|---|---|---|
| | | Site*year | 2 | 2.06 | 0.152 | |
| | Mucus | Site | 2 | 0.58 | 0.570 | |
| | | Year | 1 | 2.75 | 0.113 | |
| | | Site*year | 2 | 0.71 | 0.502 | |
| | Bioerosion | Site | 2 | 1.64 | 0.220 | |
| | | Year | 1 | 0.86 | 0.364 | |
| | | Site*year | 2 | 0.99 | 0.391 | |
| *Acropora* | Total | Site | 2 | 1.92 | 0.171 | |
| | | Year | 1 | 0.22 | 0.644 | |
| | | Site*year | 2 | 0.14 | 0.872 | |
| | Bleaching | Site | 2 | 1.27 | 0.300 | |
| | | Year | 1 | 1.02 | 0.323 | |
| | | Site*year | 2 | 0.64 | 0.538 | |

**Note:**
If impaired health result is missing then it was not observed for that coral genus. Sites: EG, Eves Garden, AG, Anenomes Garden, SW, Siwa: Seasons: 2016, 2017. Bold text indicates significant difference.

lower bleaching and pigmentation (Fig. S7), supporting our previous results. Of the sediment drivers, the BEST model included both silt and the course sediments, which explained 18% of the variations in coral health in 2017. Higher sediment trap accumulation rates, although not statistically significant ($p = 0.06$; Table 5), explained 7% of the variation in health, and were most often associated with higher prevalence of pigmentation, bioerosion and bleaching (Fig. S9).

# DISCUSSION

The three reef sites in the MSCRNP are characterized by healthy coral cover yet low coral diversity. Average live coral cover among the three reefs was 30%, ranging from 22% at EG to 39% at SW. This is lower than reefs to the north in Sabah, with reports of live coral cover from 23 to 75% (*Pilcher & Cabanban, 2000*; *Chou & Tun, 2002*; *Lee, 2007*; *Praveena, Siraj & Aris, 2012*; *Waheed et al., 2016*), but greater than the average coral cover for the wider Pacific region, estimated at 22% in 2003 (*Bruno & Selig, 2007*). Previous assessment of coral cover in the early 2000s on the Miri's reefs range from 28% (*Pilcher & Cabanban, 2000*) to 22–58% (*Elcee Instrumentation Sdn Bhd, 2002*). Although the higher coral cover reported by the latter study is most likely an artefact of the methodology used (ex-situ Acoustic Ground Discrimination System), which can result in the misidentification and, therefore, quantification of live coral cover. Regardless, our data suggest that coral cover at Miri's reefs has been relatively stable over the last two decades. Miri's coral cover is comparable to both turbid and clear-water reefs (*Roy & Smith, 1971*; *Loya, 1976*; *Larcombe, Costen & Woolfe, 2001*; *Wesseling et al., 2001*; *Palmer et al., 2010*; *Goodkin et al., 2011*), yet diversity was comparatively low (14–25 genera per reef) for the Coral Triangle region. *Turak & DeVantier (2010)* reported 391 coral species (~70 genera) on reefs near Brunei (~80 km from Miri), and *Teh & Cabanban (2007)* reported 120 species within 71

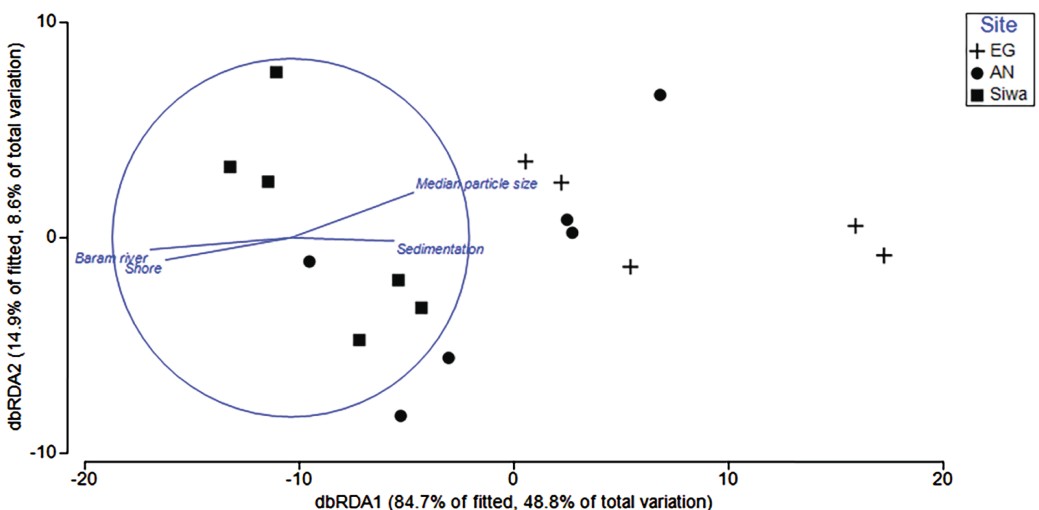

**Figure 11 Distance-based redundancy analysis (dbRDA) plot with an AIC criterion selection illustrating the significant environmental factors ($p < 0.05$) that influence community composition at EG, AG and SW.** The length and direction of the vectors represent the strength of the correlation (circle denotes a correlation of 1) and direction (+/−) of the relationship with transects (points plotted) at each site.

**Table 4 PERMANOVA results highlighting the significant drivers that explain variation in benthic community assemblage among reefs in 2017.**

| Explanatory variable | $p$ value | Pseudo-$F$ | $R^2$ |
|---|---|---|---|
| Depth | 0.094 | 2.3 | 0.010 |
| Dist. Baram River | *0.002* | 7.0 | 0.303 |
| Dist. Shore | *0.007* | 5.1 | 0.008 |
| Sedimentation rate | *0.025* | 3.9 | 0.023 |
| Course sediments | 0.069 | 2.7 | 0.001 |
| Fine sediments | 0.070 | 2.7 | 0.100 |
| Silt | 0.153 | 1.9 | 0.015 |
| Median particle size | *0.010* | 5.0 | 0.164 |

**Note:**
Bold text indicates significant difference.

hard coral genera for Banggi Island in Sabah. A comprehensive biodiversity assessment of all 30 reefs with the MSCRNP in 2001 reported 66 genera (203 coral species; *Elcee Instrumentation Sdn Bhd, 2002*). We only observed a third of the number of coral genera ($n = 28$), which is expected given we surveyed only 10% ($n = 3$) of the reefs surveyed in 2001. This report also found that coral diversity was highly variable among reefs, with an average of nine coral genera per transect. It is likely that MSCRNP reefs found further to the south and in deeper (15–35 m) offshore waters but outside the scope of this study (characterized by different environmental conditions) include several coral species not observed at our shallow nearshore sites, which are influenced by terrestrial sedimentation from both natural and anthropogenic processes.
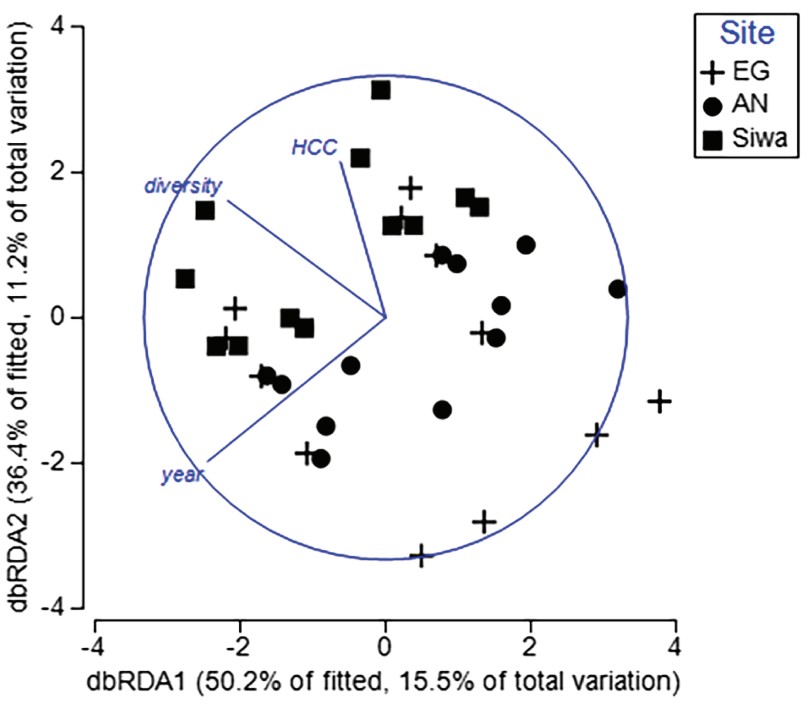

**Figure 12 Distance-based redundancy analysis (dbRDA) plot with an AIC criterion selection illustrating the influence of coral health at EG, AG and SW.** Significant explanatory variables ($p <$ 0.05; HCC = hard coral cover, diversity = coral diversity, year = Sept 2016 and May 2017). The length and direction of the vectors represent the strength and direction (+/−) of the relationship with transects (points plotted) at each site. (Image credit: Amitay Moody).

Low diversity at the surveyed sites is likely the result of poor water quality in the nearshore shallow coastal zone. The inshore reefs of Miri are found in a narrow depth range between 7 and 15 m, hence there is a complete lack of reef structure at 1–5 m. These very shallow depths, however, are often characterized by a distinct set of coral species (*Morgan et al., 2016*; *DeVantier & Turak, 2017*). Given that lack of reef structure at these shallow depths in Miri, this in part may explain lower coral diversity than on reefs to the north in Brunei and Sabah that have reached sea level. But these inshore reefs are also characterized by high levels of terrigenous sediments, which can also reduce coral diversity (*Rogers, 1990*; *Fabricius, 2005*). High sediment input from rivers is typically correlated with high nutrient loads that can lead to an increase in reef algal biomass (*De'ath et al., 2012*). Algal cover on all three reefs was high (>50%) compared to reefs in northern Borneo (0–29%; *Waheed et al., 2016*), and will most likely be competing with corals for reef space. Some coral taxa will be less resilient to both sediments and algal competition resulting in lower coral diversity (*Fabricius et al., 2005*; *De'ath & Fabricius, 2010*). In Indonesia, *Edinger et al. (1998)*, recorded lowest coral diversity on reefs with algae cover reaching 46%. Reduced diversity was also attributed to land pollution as well as destructive and over-fishing practices that destroy the reef structure and reduce fish biomass thereby removing the top-down control on algal growth by herbivore browsers (*Hughes, 1994*; *Rogers & Miller, 2006*; *De'ath & Fabricius, 2010*). In Miri, overfishing and poor land management practices have been a long-term concern for the regional government

**Table 5 PERMANOVA results highlighting the significant drivers in coral health.**

|  | *p* value | Pseudo-*F* | $R^2$ |
|---|---|---|---|
| **Explanatory variable** |  |  |  |
| Year | *0.003* | 5.0 | 0.128 |
| HCC | *0.042* | 2.8 | 0.052 |
| Diversity | *0.003* | 5.1 | 0.129 |
| Dist. Baram River | 0.304 | 1.3 | 0.019 |
| Dist. Shore | 0.521 | 0.8 | 0.020 |
| Depth | 0.467 | 0.9 | 0.017 |
| **Sediment variable** |  |  |  |
| Sedimentation rate | 0.059 | 2.4 | 0.070 |
| Course sediments | *0.031* | 2.9 | 0.152 |
| Fine sediments | *0.031* | 2.9 | 0.030 |
| Silt | 0.067 | 2.3 | 0.110 |
| Median particle size | 0.083 | 2.2 | 0.024 |

Note:
The top panel are the results of a DistLM that includes substrate structure and physical conditions among reefs and across both sampling seasons, and the bottom panel are the results of a DistLM that includes data from the sediment traps among reefs in 2017 only. Bold text indicates significant difference.

(*Elcee Instrumentation Sdn Bhd, 2002*) but there are limited funds to actively protect the reefs (*Teh & Teh, 2014*) and collect data on these impacts.

Low coral diversity does not necessarily suggest a degraded reef condition. Typically, low diversity in nature results in lower resilience (*Raymundo et al., 2005*) and community stability (*Bellwood et al., 2004*). Yet there is growing evidence to suggest that a few but tolerant species can maintain reef resilience to local and global impacts, and implies that the diversity-resilience links need further investigation (*Bellwood et al., 2004*; *Fabricius et al., 2005*; *Nyström et al., 2008*). A recent study on relatively undisturbed and well-protected reefs in the Philippines that looked to identify site-specific benchmarks for coral diversity, measured high coral cover (>30%) at the majority of sites, but low generic diversity (10–25 coral genera per 75 m by 25 m area; *Licuanan et al., 2017*). This highlights that high diversity is not necessarily a key benchmark for a healthy reef system. As well as assessing diversity on a reef, it is important to determine if and how coral community structure has changed over time. Significant shifts in coral composition can affect the reef's ecological function such as framework building, habitat complexity and food source diversity (*Aronson et al., 2004*; *Pratchett, 2005*; *Graham et al., 2006*). At six reef sites on the Great Barrier Reef (GBR) coral communities shifted over 12 years towards a high abundance of *Porites* spp. and soft corals; a community assemblage that is less likely to re-establish to the pre-disturbance coral assemblage (*Johns, Osborne & Logan, 2014*). Inshore reefs in Miri are similarly dominated by massive corals including *Porites* sp. and *Diploastrea* sp., with some (e.g. EG) also characterized by high soft coral cover (>10%). While we have no long-term data sets to evaluate change in both diversity and composition, Miri's reefs may have experienced a community shift due to reduction in water quality. Yet our tendency as coral reef ecologists to focus on coral cover, composition

and diversity, has resulted in a misconception as to what constitutes an overall healthy reef (*Vroom, 2011*). Some reefs may naturally be dominated by non-coral organisms, such as calcifying algae that are equally important for reef accretion and stability, but possibly less resilient to climate change. Thus our perception of the reefs current state and its future trajectory are likely inaccurate and need adjusting to go beyond diversity assessments.

The MSCRNP reef community can best be described as representative of turbid reefs in the Indo-Pacific. The dominant coral species include several genera (*Acropora*, *Montipora*, *Porites*, *Pachyseris*, *Favites* and *Galaxea* spp.) that have been observed on nearshore reefs in Singapore (*Chou, 1988*; *Dikou & van Woesik, 2006*), the GBR (*Ayling & Ayling, 1991*; *Larcombe, Costen & Woolfe, 2001*; *Browne, Smithers & Perry, 2010*; *Morgan et al., 2016*), Thailand (*Tudhope & Scoffin, 1994*), Hong Kong (*Goodkin et al., 2011*) and Sabah (*Pilcher & Cabanban, 2000*). These corals are considered to be more resilient to sediment influx either through: (1) enhanced photo-acclamatory abilities required during periods of low light (e.g. *Stylophora*; *Dubinsky et al., 1984*; *Browne et al., 2014*), (2) active sediment removal processes by the coral polyp (e.g. *Goniastrea*; *Rogers, 1990*; *Erftemeijer et al., 2012*), (3) enhanced mucus production to remove settled sediments (e.g. *Porites*; *Bessell-Browne et al., 2017*) or, (4) morphological advantages that result in greater degree of vertical growth thereby reducing tissue mortality from sediment burial (e.g. *Acropora* and *Montipora*; *Erftemeijer et al., 2012*). There were also distinct differences in the community assemblages particularly between SW and EG. Siwa Reef had a mixed assemblage of branching, foliose and massive corals, whereas EG was dominated by massive corals, such as *Porites* sp. and *Diploastrea* sp. These coral community differences suggest there are significant differences in environmental drivers (including sediments) over a comparatively small spatial scale (10 km).

The inshore to offshore gradient in HCC and diversity, and differences in coral composition is heavily influenced by the spatial differences in sediment-related parameters. Over 62% of the variation in benthic cover at our three reef sites is explained by differences in depth, sediment trap accumulation rates and distance from sediment sources as well as sediment particle size characteristics. Consequently, we saw a significant increase in both coral cover and diversity with increasing distance from the river mouths. Similar observations have been reported from Indonesia and Puerto Rico, where HCC nearly halved towards shore (*Loya, 1976*; *Edinger et al., 2000*), and in Hong Kong, where inshore coral cover was 20% lower than offshore (*Goodkin et al., 2011*). Reduced coral cover occurs because of low larval recruitment as a consequence of limited hard substrate following sediment settling (*Birrell, McCook & Willis, 2005*; *Fabricius, 2005*; *Dikou & van Woesik, 2006*), colony mortality caused by anoxic conditions that occur under sediment layers (*Rogers, 1983*; *Riegl & Branch, 1995*; *Wesseling et al., 2001*), or lower coral growth as a result of limited light or more energy used for sediment clearing (*Browne, 2012*). The sediment particle size and source (marine versus terrestrial) are considered equally important to sediment volume in assessing the impacts of sediments on coral health (*Weber, Lott & Fabricius, 2006*). A recent study demonstrated that as the percentage of terrestrial sediments increases, there are greater declines in coral cover (*Bégin et al., 2016*). The significantly lower HCC and diversity at EG than at SW could be driven in part by a

higher volume of terrestrial sediments from the Baram and Miri rivers. Although we did not assess sediment origin, sediment trap accumulation rates at EG were over double that at AG and SW, which may be due to the reefs closer proximity to the two river mouths. However, it could also be the result of increased sediment resuspension in shallow water or a combination of these factors. Sediment traps do not provide a comprehensive assessment of sediment dynamics on reefs, and given that our traps were out for 7 months, we recognize that our monthly sediment trap accumulation rates can only be compared among our study sites and not to other studies. Regardless, it is likely that river flow and sediments are influencing reef health, but these reefs appear to be in a temporally stable state given low recently dead coral cover (4.35%) and the limited decline in coral cover over the last two decades.

The prevalence of impaired health signs was dominated by bioerosion and pigmentation response with no signs of coral disease (with one exception). These health indicators are typically related to high sediment and nutrient influx. High levels of bioerosion in particular has been linked to land-based pollution whereby lower light, from high turbidity, reduces $CaCO_3$ density (*Risk & Sammarco, 1991*; *Lough & Barnes, 1992*) weakening the coral skeleton and increasing susceptibility to bioeroders (e.g. molluscs, worms etc.; *Prouty et al., 2017*). Furthermore, even modest increases in nutrient levels can lead to an increase in the abundance of bioeroding organisms shifting a reef community from one of net production to net erosion (*Hallock & Schlager, 1986*; *Hallock, 1988*; *Prouty et al., 2017*). Bioerosion levels were significantly greater following the wet season when the impact of sediments on the Miri's reefs were elevated as indicated by declines in light and higher suspended sediment loads. Conversely, pigmentation rates were higher following the dry season. Pigmentation is an indicator of immune function in response to a stressor (*Willis, Page & Dinsdale, 2004*; *Palmer, Roth & Gates, 2009*). These stressors have been related to settling sediments (*Pollock et al., 2014*) or lesions from abrasion or scars (*Willis, Page & Dinsdale, 2004*), or for the case of Miri's reefs elevated SST recorded in the region in 2016 leading to the moderate bleaching event as observed by the diving operators and fisherman. Spatially, pigmentation rates were significantly lower at Siwa, which may suggest that corals at the least sediment impacted site were also less stressed than at AG and EG. Sediments can also promote diseases in corals (*Voss & Richardson, 2006*; *Haapkylä et al., 2011*; *Pollock et al., 2014*) with Black Band Disease and White Plague widely observed in the Indo-Pacific (*Harvell et al., 2007*; *Beeden et al., 2008*), although generally low (~8% of current global records) in SE Asian reefs compared to the Caribbean (*Green & Bruckner, 2000*). Suggested explanations for this include poor reporting of coral diseases and relatively high coral diversity that might aid in diminishing a quick spread of a disease (*Raymundo et al., 2005*). At Miri, the more likely explanation of low to no coral diseases are more resilient individual corals and coral species, and potentially limited connectivity with nearby coral populations, although this remains speculative until further work is carried out.

Hard coral cover and diversity also explained a significant portion of the variation in coral health. Miri's reefs with a higher frequency of impaired health at sites recorded less coral cover and diversity. In a recent study by *Miller et al. (2015)* reefs in Sabah had four

common coral diseases at varying frequencies (<0.1–0.6 per affected colonies in an m$^2$) and signs of tissue necrosis and pigmentation responses. They found a positive correlation between disease frequency and coral cover, which suggested that host density was a key driver of disease prevalence and compromised health. This relationship is due to reduced distances between colonies, and greater shading and competition by fast-growing species as coral cover increases (*Bruno & Selig, 2007*). In Miri, we see the reverse trend suggesting that factors other than host density are driving coral health, potentially changing sediment loads and finer sediment particles not present in Sabah. However, other variables often associated with sediment such as nutrient levels and pollution such as heavy metal loads are also worth investigating.

Variable species composition among sites would also partly explain the spatial variation in coral health. Different coral taxa have different susceptibilities to bioerosion, bleaching, disease and compromised health (*Raymundo et al., 2005*; *Couch et al., 2014*; *Heintz, Haapkylä & Gilbert, 2015*). In Miri signs of pigmentation and bioerosion were most prominent on massive *Porites* sp. colonies. *Porites* sp., although typically considered a hardier coral taxa (*Raymundo et al., 2005*) tolerant of turbid waters, often have the most lesions, highest tissue loss and pigmentation response (*Tribollet, Aeby & Work, 2011*; *Pollock et al., 2014*; *Heintz, Haapkylä & Gilbert, 2015*) as well as being a target for disease (*Raymundo et al., 2005*). The level of bleaching observed in *Porites* at Miri was comparable to other abundant coral genera, but recovery potential after 9 months was lower, possibly due to other stress symptoms. Bleaching was the most common sign of impaired health among coral taxa, most commonly observed in *Pachyseris*, *Porites*, *Montipora*, *Dipsastrea* and *Acropora* spp. (in declining order). A comprehensive study by *Marshall & Baird (2000)* of 40 coral taxa on the GBR found the same coral taxa were highly (>50% bleached or dead) or severely (>15% dead) susceptible to thermal stress. In contrast, the other five most abundant corals at Miri (*Diploastrea*, *Favites*, *Galaxea*, *Echinopora* and *Merulina* spp.) are considered to be less sensitive to rising SST (*Marshall & Baird, 2000*; *Guest et al., 2016*). However, bleaching susceptibility does vary considerably according to the thermal history of a region. For example, *Acropora* sp. is susceptible to bleaching on some reefs (*Marshall & Baird, 2000*; *Pratchett et al., 2013*; *Hoogenboom et al., 2017*), but is less susceptible on other reefs (e.g. Singapore following the 2010 bleaching event: *Guest et al., 2012*). Only ~5% of *Acropora* sp. colonies in Miri showed signs of thermal stress, which suggests moderate thermal tolerance to high SST. High levels of algal density in coral tissue are also linked to higher thermal stress resistance (*Glynn, 1993*; *Stimson, Sakai & Sembali, 2002*) due to the symbionts providing a greater concentration of mycosporine-like amino acids that protect corals from UV radiation (*Xu et al., 2017*). Symbiont densities measured at Miri were high (mean = 2.4 * 10$^6$ cells per cm$^2$) but comparable to corals on other turbid reefs like those from Singapore (e.g. 0.5–3 * 10$^6$ cells per cm$^2$ (*Browne et al., 2015*). However, it was *Acropora* sp. that had significantly higher symbiont density than the more frequently bleached *Montipora* sp. and *Pachyseris* sp. Our results suggest that resilience to stress for these corals is a complex, but synergistic relationship between the level and frequency of environmental stressors, community composition and a coral's adaptability to increased SST.

In 2016, a severe coral bleaching event occurred throughout the Indo-Pacific region. The impacts of this event were thoroughly assessed on the GBR, where over 90% of reefs bleached resulting in the loss of 29% of shallow-water coral cover (*Great Barrier Reef Marine Park Authority, 2016*). In January to March 2016, SST along the northern shore of Borneo were in the highest 10% of global records since 1990 (*Great Barrier Reef Marine Park Authority, 2016*). Sea surface temperatures reported by NOAA for Brunei peaked in May to June 2016 at 31 °C (the bleaching threshold temperature; Fig. 12). During this time there was 1–2.5 Degree Heating Weeks and mid-level bleaching warnings. Sea surface temperatures remained at ~30 °C until January 2017 (*National Oceanic & Atmospheric Administration, 2018*), which agree with our in-water assessment of SST during September 2016 to early 2017 (Fig. S2). This suggests that while corals at Miri were subject to elevated SST for five or more months, our surveys revealed comparatively low bleaching rates (~10% of colonies bleached), and high recovery rates (as suggested by the tagged corals; >90%). This suggests that these nearshore turbid water reefs are resilient to high SST supporting the growing body of evidence that turbid reefs bleach less severely and frequently than their clear-water counterparts (*Marshall & Baird, 2000*; *Heintz, Haapkylä & Gilbert, 2015*; *Morgan et al., 2017*). Low bleaching and high recovery rates of Miri's reefs are possibly due to nearshore coral assemblages more frequent exposure to higher temperatures than their offshore deeper conspecifics, resulting in the development of adaptive mechanisms (*Marshall & Baird, 2000*; *Guinotte, Buddemeier & Kleypas, 2003*; *Guest et al., 2016*; *Morgan et al., 2017*). It may also be due to lower UV light penetration that can exacerbate temperature stress (*Courtial et al., 2017*), or potentially from higher heterotrophy, which increases the supply of essential metals to the symbionts thus sustaining them through elevated temperatures (*Ferrier-Pagès, Sauzéat & Balter, 2018*). This study further suggests that while turbid reefs are potentially more resilient to elevated SST, the mechanism/s responsible for this resilience remain unclear.

## CONCLUSIONS

In conclusion, the MSCRNP reefs are characterized by relatively high coral cover, low prevalence of impaired health and are composed of a few but tolerant coral taxa. Low recently dead coral cover and potentially limited decline in coral cover over the last two decades indicate these reefs are stable despite elevated sediment inputs and regular exposure to thermal stress events. There are, however, potential risks from proposed coastal and in-land developments given we found that sediment-related parameters have resulted in an on- to offshore gradient in coral cover, diversity and health. Furthermore, high bioerosion and algae cover indirectly suggests high nutrient influx, most likely from the Baram River. The high prevalence of bioerosion observed in *Porites* sp. colonies is a concern given that this coral is a key reef framework builder, and any notable declines in *Porites* sp. health will reduce coral reef complexity and habitat availability for other invertebrate and fish species. Currently, there is no baseline data on spatial and temporal changes in river outputs and sediment plume dynamics within the MSCRNP, which is crucial in evaluating future threats to these reefs. Local management agencies will need to address this knowledge gap if they plan to develop strategies that address the potential

impacts of changing land use on MSCRNP. The reefs current health state and elevated stress tolerance does, however give hope that these reefs could be resilient to future climate change but only if local water quality does not deteriorate.

## ACKNOWLEDGEMENTS

We would like to thank the Curtin Malaysia Research Institute for facilitating this research, especially Prof Clem Kuek and Ms Daisy Saban who worked tirelessly to make sure our research trips went to plan. We thank our volunteers Amitay Moody, Hedwig Krawczyk (who also produced Figs. 1 and 12), Paula Cartwright, Toloy Keripin Munsang, Valentino Anak Jempo, Sun Veer and the numerous volunteers from the Curtin Sarawak Dive Club for their assistance with field work. Thanks also to the captains and crew from Coco Dive Centre and Hoopa Dive.

### Funding

This work was supported by the Curtin Malaysia Research Institute, the National Geographic Society Research Grant (No. CP 025ER 17) and the German Academic Exchange Service (No. 57318354). Jens Zinke was supported by a Royal Society Wolfson Fellowship (RSWF\FT\180000). Jens Zinke and Christina Braoun were supported by a German Academic Exchange Agency (DAAD) grant (57318354). Christina Braoun was supported by the Ernst Reuter Stiftung of the FU Berlin. Christina Braoun also received technical support from the Aquatech Group, UK who provided the Aqualogger. The funders had no role in study design, data collection and analysis, decision to publish, or preparation of the manuscript.

### Grant Disclosures

The following grant information was disclosed by the authors:
Curtin Malaysia Research Institute.
German Acdemic Exchange Service: 57318354.
Royal Society Wolfson Fellowship: RSWF\FT\180000.
German Academic Exchange Agency (DAAD) grant: 57318354.
Ernst Reuter Stiftung of the FU Berlin.
Aquatech Group, UK.

### Competing Interests

The authors declare that they have no competing interests.

### Author Contributions

- Nicola Browne conceived and designed the experiments, performed the experiments, analyzed the data, contributed reagents/materials/analysis tools, prepared figures and/or tables, authored or reviewed drafts of the paper, approved the final draft.

- Christina Braoun performed the experiments, analyzed the data, prepared figures and/or tables, authored or reviewed drafts of the paper, approved the final draft.
- Jennifer McIlwain performed the experiments, prepared figures and/or tables, authored or reviewed drafts of the paper, approved the final draft.
- Ramasamy Nagarajan analyzed the data, contributed reagents/materials/analysis tools, authored or reviewed drafts of the paper.
- Jens Zinke contributed reagents/materials/analysis tools, prepared figures and/or tables, authored or reviewed drafts of the paper, approved the final draft.

### Field Study Permissions

The following information was supplied relating to field study approvals (i.e., approving body and any reference numbers):

Field experiments were approved by the Sarawak Forestry Commission (permit no. (61) JHS/NCCD/600-7/2/107).

### Data Availability

Raw data is available in the Supplemental Files.

### Supplemental Information

Supplemental information for this article can be found online at http://dx.doi.org/10.7717/peerj.7382#supplemental-information.

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
