# Peer review of "Borneo coral reefs subject to high sediment loads show evidence of resilience to various environmental stressors"

_PeerJ, doi:10.7717/peerj.7382_

## Round 0.1 · original submission · Major Revisions

Three expert reviewers have evaluated your manuscript entitled ­"Inshore, turbid coral reefs from northwest Borneo exhibiting low diversity, but high cover show evidence of resilience to various environmental stressors" and their reviews are attached. The topic is interesting and under-studied, however there are many important issues that have been pointed out. Based on these reviews I cannot recommend that your manuscript, in its current form, be accepted. However, I do encourage a resubmission of this manuscript ensuring that the reviewer´s concerns are addressed and I am positive that the comments from all three reviewers will prove very useful to you as your prepare your manuscript for further consideration. I cannot stress strongly enough how important it is for you to answer and justify your response to every comment that has been made. In addition, when adding or deleting text, please ensure to refer to the line numbers in the original manuscript as well as the line number of the new manuscript where appropriate so that all changes can be easily found and referred to.

·

Basic reporting

This manuscript presents research on a question very relevant to coral reef ecology and conservation. Specifically, the authors seek to:
“1) quantify benthic cover, coral cover and health, 2) compare the prevalence of impaired health in the dominant coral species, 3) identify key parameters related to sediment delivery that influence benthic cover and health along an inshore to offshore gradient, and 4) assess how resilient these inshore reefs are to future changes in sediment supply.”
They present results relevant to these questions, though I have some questions about the collection and interpretation of some of the environmental data—see section on experimental design.
The manuscript is generally well written with clear sentences. There are a few acronyms that were not defined the first time they were used. Appropriate references were used to support most points but there are one or two instances where a key reference was missed (in particular, Storlazzi 2011—see comments in pdf) and several instances where a citation was used incorrectly (e.g. were listed when the reference did not support the statement made in the manuscript). Several references are incomplete in the reference section.
There are several instances in which sentences needs to be clarified, see pdf for specific comments.
Figures are very helpful, but:
Fig. 1: add a box around the Miri area on the small-scale map, to show the area that is enlarged.
Fig. 2: state clearly in the legend which sites these environmental data come from, which is important since you did not measure these parameters at all sites
Fig. 3: change sedimentation rate to “trap accumulation rate” or “trap sedimentation rate”, or other wording to highlight that these measures are from sediment traps, which don’t reflect actual accumulation rate on the benthos
Fig. 4: It would be worthwhile to see the variation in sediment size for each site. As it is presented I’m guessing that we are seeing the mean for all traps at a given site (should specify in legend). Possibly instead of showing all size fraction, show % mud (mean + SE) for each site
Fig. 5: Same comment as Fig. 4 about not seeing a measure of variation
Fig. 6: it is difficult to see feeding scars in 6B. There is no explanation of what is shown in 6D. explain what non-focal bleaching represents.
Fig. 7: Indicate what the error bars represent. Standard error?
Fig. 8: same comment as Fig. 7.
Fig. 9: same comment as figure 5.
Fig. 10: same comment as Figs. 7-8
Fig. 11: consider providing a more thorough explanation of the figure in the legend. Explain the Eigenvectors and what the length and direction means. Explain what the circle represents. Should the legend read “Distance-based redundancy analysis plot” rather than DistLM plot? I would recommend removing the box that automatically comes in PRIMER which highlights the transformation and resemblance analysis; best to put this in the legend.
Fig 12: Same comments as fig. 11

Experimental design

This manuscript presents research in coral reef ecology, which is within the aims and scope of Peer-J (Biological Sciences, Environmental Sciences). It sets the stage clearly for the need for this research (i.e. better understand the dynamics of turbid reefs, and assessing their resilience to future disturbances).

The research questions are defined clearly at the end of the introduction, and the manuscript presents data to answer these questions. Methods are a bit unclear at times, and in some cases some weakness must be addressed (at least in the discussion and interpretation of results should be made in light of these weaknesses).
Line 142: Explain what “odyssey and HOBO” mean.
Line 143: explain why these parameters were only recorded at one site, and implications for extrapolation of these data
Line 144: I worry about how effective light-measuring devices are after 2 weeks in the field—would they get covered by sediment over that time period?
Line 148: weather was found on worldwideweatheronline.com—which site?
Line 153: look up Storlazzi 2011 use and misuse of sediment traps in coral reef environments (journal coral reefs) for a thorough discussion of these traps and what they measure. Did yours include baffles?
Line 154: sediment traps are usually changed much more frequently than 9 months. I worry about organisms settling in the traps over this long time period and influencing the amount/type of sediment collected in the end.
Line 163 (and many other places in the manuscript): refer to “trap accumulation rate” rather than “sediment accumulation rate”, since traps overestimate actual accumulation rate
Line 166: describe in greater details how sediments were homogenized.
Line 167: provide the range of size for each size class
Line 176: explain if you used the entire benthic photos as the “quadrat” that you then analyze with CPCe. How big is this quadrat?
Line 177: are those points completely random, or stratified in some way?
183: how did you quantify bioerosion?
187: define focal and non-focal bleaching
195: coral genera, not species. Why these genera only and not Porites?
208: clarify your analysis and unit of replication
217: unclear why you used a Kruskal-Wallis
226: why did you lump all photosynthetic organisms in algae? At least in the Caribbean it makes some sense to divide into macroalgae, turf algae and filamentous cyanobacteria. It seems odd to me to include H’ and richness in the community assemblage.
231: why Euclidean rather than Bray-Curtis in this case?
241: this statement needs a reference. I’m not sure it’s something that occurs broadly, and across a wide range of % coral cover
242: But couldn't one argue this point the other way? What if a reef has a very low diversity, all with species that are quite susceptible to disease?

More generally on the methods:
• sediment traps have been used a lot traditionally, but recent work has really highlighting their limitations in coral reef environments. You should reference the Storlazzi et al 2011 paper and make sure that your interpretation of sediment data takes into consideration what they highlight.
• Your results present chi-square analyses but these are not described in the methods. Clarify and justify why a chi-square analysis.

Validity of the findings

The study provides new and valuable data on the status of coral reefs in this region.

I have questions about some analyses, including why chi-square analyses were performed in what seems better analyzed with an ANOVA (continuous dependent variable). The discussion could be fleshed out in several places, and in some cases a point needs to be reworded so that it accurately reflects the references cited, or to keep the statement within the scope of what can be said based of the data provided.

There are misinterpreted/misreported references in these lines: 396, 448, 463

Please review the first section of the results in Begin et al 2013. Sediment size and composition are not related in this study. Amend your statements that make broad generalizations suggesting that sediment size can be used as an indication of sediment origin.
Similarly, be careful of your statements suggesting that terrigenous sediment always have more organic material, or that the organic material is the cause of their negative impact.

When stating that cover of any benthic component is high or low, try to provide data from other reefs to compare. Similarly when comparing what is found on these inshore turbid reefs (and speculation on the processes causing these patterns) compared to deeper offshore reefs.

Be very careful when comparing sedimentation rate measured here via traps to sedimentation rate reported in other studies, in particular if those rates were measure by other methods. Make sure that you clearly highlight that your results represent trap accumulation rates.

Lines 433-435: reword. This is too strong a statement. You are finding correlation, and don’t have the data necessary to state causation. Additionally, you list “depth” as a “sediment-related factor”, which is arguable.

Generally, the discussion needs to be fleshed out to better explore the various potential causes for the patterns observed here and be more nuanced about the limitations of the study.

Line 283-4: clarify is this is H’ rather than richness
Line 286: was a quantitative analysis performed about this? Provide results.
Line 345: There is no Fig 11a

See pdf for more in-text comments

Additional comments

This study is worthwhile and provides data on reefs in an under-studied area with limited baseline data. It also measures multiple environmental parameters that are related to the study questions, which strengthens the arguments and conclusions. However the limitations of the data need to be made more clear (e.g. that some environmental data are only from some sites; better contextualize the sediment trap accumulations rates), and the authors need to revisit some assumptions (e.g. that fine sediment is of terrestrial origin). More details are needed in the methods and the discussion needs to be fleshed out some more. The authors should also review all references and make sure that they are appropriate as citations for the statement they are cited in. I found several problems on that level and wonder if there are more that I didn’t catch. This study is a worthwhile one that should be published, but it does need more polishing to address the current weaknesses.

Reviewer 2 ·

Basic reporting

The manuscript is generally well written. There are minor formatting issues (scientific names are not italicized, the “sp.” after a generic epithet should not be italicized)

The authors cite several studies but I suggest the following key papers in the relevant literature should be included:

Darling, E.S., Alvarez‐Filip, L., Oliver, T.A., McClanahan, T.R. and Côté, I.M., 2012. Evaluating life‐history strategies of reef corals from species traits. Ecology Letters, 15(12), pp.1378-1386.

DeVantier, L.M., De’Ath, G., Turak, E., Done, T.J. and Fabricius, K.E., 2006. Species richness and community structure of reef-building corals on the nearshore Great Barrier Reef. Coral reefs, 25(3), pp.329-340.

DeVantier, L. and Turak, E., 2017. Species richness and relative abundance of reef-building corals in the Indo-West Pacific. Diversity, 9(3), p.25.

Done, T., Turak, E., Wakeford, M., DeVantier, L., McDonald, A. and Fisk, D., 2007. Decadal changes in turbid-water coral communities at Pandora Reef: loss of resilience or too soon to tell?. Coral Reefs, 26(4), pp.789-805.

Flower, J., Ortiz, J.C., Chollett, I., Abdullah, S., Castro-Sanguino, C., Hock, K., Lam, V. and Mumby, P.J., 2017. Interpreting coral reef monitoring data: A guide for improved management decisions. Ecological indicators, 72, pp.848-869.

Johns, K.A., Osborne, K.O. and Logan, M., 2014. Contrasting rates of coral recovery and reassembly in coral communities on the Great Barrier Reef. Coral Reefs, 33(3), pp.553-563.

Licuanan, W.Y., Robles, R., Dygico, M., Songco, A. and van Woesik, R., 2017. Coral benchmarks in the center of biodiversity. Marine pollution bulletin, 114(2), pp.1135-1140.

Smith, J.E., Brainard, R., Carter, A., Grillo, S., Edwards, C., Harris, J., Lewis, L., Obura, D., Rohwer, F., Sala, E. and Vroom, P.S., 2016. Re-evaluating the health of coral reef communities: baselines and evidence for human impacts across the central Pacific. Proc. R. Soc. B, 283(1822), p.20151985.

Vroom, P.S., 2011. “Coral dominance”: a dangerous ecosystem misnomer?. Journal of Marine Biology, 2011.

I suggest assigning numbers to sites (e.g., Eve’s Garden as Site 1, Anemone Garden as Site 2, Siwa Reef as Site 3. Site numbers related to distance are easier to remember than acronyms.

I suggest the authors organize their findings and discussion better to substantiate the conclusion that the reefs they study are resilient

Experimental design

Please specify the sampling domain and how the domain was sampled to allow us to determine to where the inferences made could be applied. In particular, please state where in the reef the transects were positioned (i.e., leeward or windward relative to the monsoons of SE Asia; the reef zone sampled (upper reef slope? lagoon?) and identify which are the replicates and which are the subsamples. Was there randomization or was sampling haphazard? What is the basis for determining transect length used?

Please show a more detailed map showing the reef outlines and where in the reef the transects were located.

The methods section identified the use of the Kruskal Wallis test. Are these the many instances of the Chi-square values being reported?

Please clarify the substrate categories (particularly algae on rock, algae on rubble, and algae on coral) used with CPCe. Why was this not discussed in the results? Was turf algae lumped with macro-algae? Note that short turf algae is available for coral recruitment (Flower et al 2017).

Please describe in detail how disease, bleaching, bioerosion, pigmentation, mucus production, scars were quantified.

Validity of the findings

I suggest the authors organize their findings and discussion better to substantiate the conclusion that the reefs they study are resilient

Regarding lines 365-375 of the discussion, the papers listed earlier (particularly Smith, Vroom, Licuanan) state coral cover gives indications on suitability of a reef to coral growth but not resilience. The conclusion concerning the latter need to be substantiated. Personally I think the lower diversity found (lines 382-383) indicates loss of resilience in the Miri reefs studied. There might have been recovery but not reassembly (using the terminology of Johns et al). Note that many of the dominant coral genera reported are stress tolerators (see the paper by Darling et al). Are the Porites listed branching or massive? The latter paper indicates these two forms of the genus have different life history strategies. Lines 426-431 indicate life history patterns are more telling than hard coral cover. However, valid comparisons can be made only if the same reef zones (depth, leeward or windward sides of reefs) and reef habitats (reef slope, lagoon, etc) are being compared.

Please explain what lack of structure (line 392) means. It appears the reef communities being compared are not comparable given the differences in depth and the absence of information on the locations of the transects.

The authors’ statements about the stability of coral cover (lines 374 to 375) are rather speculative given the typical spatial (within and between) variability of hard coral cover. Ideally the authors demonstrate that the previous studies cited looked at the same depths, the same zone of the reefs, and the same reefs.

Reviewer 3 ·

Basic reporting

I have read this manuscript very carefully and I find it very difficult to understand the main message that the authors are trying to convey. I am not certain that they are using the word “resilient” in the strict sense. At times they maintain the study reefs are resilient and at other times they emphasize their susceptibility to stressors.
It does not make sense to speak of reefs as “resilient” without identifying the stressors they are resisting or re-bounding from. And they are using the word “health” without defining it
In general, I found a lack of clarity in the writing, with words used in very imprecise ways. For example, line 106, what is meant by reef “stability”. What is meant by reef “function”? What do chlorophyll a and symbiont density indicate?
In general, I think that the lack of precision in the writing may reflect a lack of understanding of some of the basic concepts and mis-interpretation of some of recent scientific literature.
Some of the literature citations do not seem to support the associated statements.

Experimental design

I found it hard to determine exactly how the research was conducted. Some of the methods are not sufficiently described.
Were the transects selected randomly? ( line 137) That is, independently of each other? It does not appear so. In that case, it is not possible to extrapolate the results to entire reef areas. The data could still be useful, but the fact that the transects were chosen haphazardly should definitely be discussed. If you can not be sure that the sampling units (transects) were representative of the reef area, you can not compare reefs. If the transects were permanent and data were collected on the two occasions from exactly the same transects, more can perhaps be said with confidence.
If the transects were not selected randomly, then at least some of the statistical analysis is not valid.
And it is not clear without a pilot study if 6 transects is a sufficient number.
Line 153--- I think there are problems leaving sediment traps out for such a long period of time. Were salts rinsed from the sediments?
Line 183—need more explanation as to how each of these signs were identified
Line 190 this is a very small sample size—explain also that overall bleaching was calculated

Validity of the findings

I am not able to evaluate the statistical analyses in the paper.
Some of the conclusions are not well-supported. In general, it appears that the authors are confusing correlation with causation in some of their statements
Line 240-- Higher coral cover is not always associated with greater probability of “impaired health”
Line 241—I think this is an oversimplification of Mydlarz et al’s paper. If you have a reef with just a few species that are susceptible to diseases, then you have a low diversity reef that is not resistant.
Line 244—requires revision
Line 245-- what are “substrate structure predictors”?
Line 252-- what units were used for PAR?
Line 257 Rogers 1990 suggested further research was needed on sedimentation thresholds and it is not correct to state that this rate, based initially on just a few studies, is a threshold suitable for describing all reefs
Line 302—how was bioerosion identified in the field?
Line 305—what is the reference/citation for a 1evel of 10% of bleaching being low?? If 10% of the coral bleached each year, and it was a different subset of colonies, that would certainly be a sign of high thermal stress
Line 340 what does this really tell us? 62.5%? not clear what this means
Line 353—so does this say that high diversity is not related to increased bleaching?
Line 365 and on – this discussion of relatively high coral cover is very confusing and not very convincing—
Line 383—once again, not convincing—any suggestion of trends in cover or biodiversity are not based on any real evidence
Once again, the language is not precise
Line 387—so if these reefs are “resilient”, why is there this suggestion that some species are missing because of high sedimentation?
Line 414—can this statement be supported?
Line 434—what is a “gradient” in coral composition?
Line 453—these sediment data cannot “confirm” anything given it is not known if they are terrestrial or not
Line 458—I don’t think 20% is a low prevalence!!
Line 472—I don’t recall a bleaching event being documented at their study sites
Line 480—I don’t understand this statement—and it may contradict what was said earlier in the paper
Line 486-487 doesn’t this also contradict earlier statements?
Line 538-- I would think that there might be some observations available for the time of peak sea water temperatures—from dive tour operators or someone? It is not appropriate to assume that there was little bleaching on these study reefs

Additional comments

I hope that the authors will be able to substantially revise this paper. I had a very difficult time trying to follow just what was being said. I think that there is some confusion regarding some of the key recent scientific references.I think that the paper would be substantially improved if the authors very carefully checked on their literature citations to ensure that they are accurately cited.
In some cases, it appears that the authors are confusing correlation with causation in some of their statements.
A few more specific comments follow:
Line 63—requires revision.
Line 85-- What does “fair” indicate? Did the Reef Check surveys measure coral cover?
If the only coral cover measurements date back to 2000 and then 2002 (with a different method), then we do not know what the trends have been over the last several years. (see related conclusion near end of the paper)
Line 116 I do not understand this statement “why turbid reefs are resilient”--
Line 124-- here it says that the study reefs had 30 to 50% cover but they cite a 2002 paper/reference—
When did the reefs have this cover? Is this what the authors measured?

---

## Round 0.2 · Minor Revisions

Two expert reviewers have evaluated your re-submission and their comments can be seen below and in an attached PDF. Both feel that the manuscript has greatly improved however there are still observations that need to be attended such as using correct literature, more details in the methods, describing the reef system you studied in more detail and placing a lot of attention to carefully proof reading and correcting mistakes. Please note that PeerJ does not offer am editing service so I encourage you to pay particular attention to this aspect.

·

Basic reporting

This is a much improved manuscript compared to the first one. However several sentences are still overstated, and there are some typos and other issues throughout. Specific comments written in pdf. Some references not entirely appropriate for the sentence they are inserted into.

Experimental design

Some more information needed in methods. One question about the actual size of quadrats, the size stated doesn't seem right. The discussion mentions non-coral reef builders. Were crustose coralline algae quantified? If so it'd be good to report. If not, explain why. More comments throughout the pdf.

Validity of the findings

look for comments in pdf. Some words/sentences need to be qualified and toned down.

Reviewer 2 ·

Basic reporting

No comment

Experimental design

I suggest the authors state in the manuscript that the reefs studied do not have the typical windward-leeward zonation patterns and are thus more homogenous. This will help readers better understand the considerations for sampling.

Validity of the findings

I suggest the authors RESTATE in the manuscript that the reefs studied are small and do not have the typical windward-leeward and depth-related zonation patterns. This will help readers better understand when and where the conclusions of the authors most likely apply

Additional comments

Some proof reading of the manuscript and its attachments are still needed. As examples: the scientific names are not properly formatted in the text e.g., (line 458: "Faviidae" should not be italicized) and in the tables (e.g., Table 1) . Line 438: "generic" rather than "genetic" diversity; Line 455: "may be" rather than "maybe"

---

## Round 0.3 · accepted · Accept

I am satisfied with the changes that have been made to the manuscript.